# Virus–Host Interaction Gets *Curiouser and Curiouser*. PART I: Phage P1*vir* Enhanced Development in an *E. coli* DksA-Deficient Cell

**DOI:** 10.3390/ijms22115890

**Published:** 2021-05-31

**Authors:** Grzegorz M. Cech, Anna Kloska, Klaudyna Krause, Katarzyna Potrykus, Michael Cashel, Agnieszka Szalewska-Pałasz

**Affiliations:** 1Department of Bacterial Molecular Genetics, University of Gdańsk, Wita Stwosza 59, 80-308 Gdańsk, Poland; klaudyna.krause@phdstud.ug.edu.pl (K.K.); katarzyna.potrykus@ug.edu.pl (K.P.); agnieszka.szalewska-palasz@ug.edu.pl (A.S.-P.); 2Department of Medical Biology and Genetics, University of Gdańsk, Wita Stwosza 59, 80-308 Gdańsk, Poland; anna.kloska@ug.edu.pl; 3Intramural Program, Eunice Kennedy Shriver Institute of Child Health and Human Development, National Institutes of Health, Bethesda, MD 20892, USA; cashelm@mail.nih.gov

**Keywords:** P1*vir* bacteriophage, P1 phage, DksA, lytic development, host-virus interactions

## Abstract

Bacteriophage P1 is among the best described bacterial viruses used in molecular biology. Here, we report that deficiency in the host cell DksA protein, an *E. coli* global transcription regulator, improves P1 lytic development. Using genetic and microbiological approaches, we investigated several aspects of P1*vir* biology in an attempt to understand the basis of this phenomenon. We found several minor improvements in phage development in the *dksA* mutant host, including more efficient adsorption to bacterial cell and phage DNA replication. In addition, gene expression of the main repressor of lysogeny C1, the late promoter activator Lpa, and lysozyme are downregulated in the *dksA* mutant. We also found nucleotide substitutions located in the phage immunity region *imm*I, which may be responsible for permanent virulence of phage P1*vir*. We suggest that downregulation of C1 may lead to a less effective repression of lysogeny maintaining genes and that P1*vir* may be balancing between lysis and lysogeny, although finally it is able to enter the lytic pathway only. The mentioned improvements, such as more efficient replication and more “gentle” cell lysis, while considered minor individually, together may account for the phenomenon of a more efficient P1 phage development in a DksA-deficient host.

## 1. Introduction

Bacteriophage P1, along with λ phage and T4, is among the best described bacterial viruses in molecular biology. The P1 features, as well as its life cycle, have been studied over many years [1], and its complete genome was published in 2004 [2]. The P1 phage ability to conduct generalized transduction made it a valuable tool in molecular biology [3], used for genetic engineering and gene transfer, a role that even nowadays should not be underestimated [4]. Moreover, P1 served as a model to study various processes in molecular biology, such as DNA replication, recombination and multiple aspects of phage-host interactions, as widely summarized in [1]. 

P1 can infect a variety of Gram-negative hosts, including many species from the *Enterobacteriaceae* family [5], and its interactions with the host were mainly studied in the *Escherichia coli* model. Upon infection and entry into the host cell, P1 genomic DNA undergoes circularization mediated by phage-encoded Cre-dependent recombination [6]. At that time, as a temperate phage, P1 makes a choice between two different developmental strategies, i.e., lysis versus lysogeny. As a lysogen, P1 genome exists in the host cell as a large extrachromosomal genetic element—a low copy number plasmid [7]. 

The lysis–lysogeny decision is mediated by an immunity system which includes three immunity regions: *imm**C, immT* and *imm*I [8]. It needs to be emphasized that the P1*vir* phage, a virulent strain used in this study, was not constructed but isolated in the late 1960s and described as harboring unknown mutation(s) in the secondary repressor region—*imm*I [9].

The choice of the phage life strategy is determined by a molecular interplay between products of genes expressed immediately upon infection. The major players encoded by the *imm*C region are the main repressor protein, C1, and the C1 repressor inactivator protein, Coi [10,11]. C1 binds to the operator sequences present in the P1 genome and represses transcription from promoters located near these operators; this is necessary for lysogeny to occur. The Coi protein binds directly to C1, blocking its function and promoting lysis. Thus, the balance between these two regulators determines the choice between lysis and lysogeny. 

In addition, the Ant protein (Ant1-Ant2 dimer), encoded in the *imm*I region, acts as a secondary anti-repressor which binds C1. Thus, for establishing lysogeny, expression of the *ant* genes has to be blocked, and this is achieved by C1-mediated transcription repression of an operon containing the *c4*, *icd* and *ant1/2* genes. However, full repression of *ant* expression also requires the action of C4 antisense RNA [12]. The *icd* gene in this operon plays a role in negative regulation of *ant* expression. Moreover, Icd inhibits host cell division which provides time needed for either establishing lysogeny or entering into the lytic pathway [13]. Yet another player in this complex regulation is the Lxc corepressor encoded by the *imm*T region. Lxc enhances the ability of C1 to bind to the mentioned P1 operators [14]. 

Once lysogeny is established, the P1 genome replicates similarly to plasmid DNA. This plasmid has two replication origins, *oriR* and *oriL*, and exhibits a complex regulation of replication, partition and addiction [1]. Replication starts at *oriR* and it is initiated with P1 encoded RepA protein. The second origin, *oriL*, is involved during lytic development. Replication from both origins proceeds via the theta model, however later in the lytic development it switches to the sigma model, which enables phage heads’ packing [15]. 

The lytic pathway can be chosen upon infection, but changing environmental conditions may also provoke P1 lysogen to enter the lytic pathway. In that case, many genes formerly repressed by C1 undergo expression, including early genes whose products are responsible for initiation of the replication process and its control. Lytic replication is initiated at *oriL* by the RepL protein. The *repL* gene is co-transcribed with *kilA* which encodes a protein that inhibits cell division during lytic phage development. Other early proteins are involved in regulation of gene expression, recombination, and methylation [1]. The late genes, including those whose products are involved in phage morphogenesis, packaging of virus particles and cell lysis, are typically expressed from specific promoter sequences whose transcription requires a phage-encoded regulator, the Lpa protein, defined as an RNA polymerase binding factor [16]. Assembled and accumulated phage particles are released from the host cell due to the action of phage-encoded lysozyme, whose function is controlled by a system of holin and antiholin proteins (LydA and LydB) [17]. 

In their development, bacteriophages rely on and are affected by multiple host regulatory processes and factors. DksA was identified as a regulatory protein affecting many processes, e.g., heat stress response [18], and its role as a transcription regulator involved in the stringent response [19,20] placed DksA among proteins intensively studied by many research groups. Its function in the regulation of various processes has been reported, including DNA repair, quorum sensing, and virulence [21,22,23]. Thus, DksA deficiency affects numerous cellular processes causing pleiotropic effects. 

We discovered that a host cell deficiency in DksA affects bacteriophage P1 life cycle leading to its improved lytic development; we attempt to understand the basis of this effect by using genetic and microbiological approaches. As the phage development is a multistep process, many factors including host features, the process of adsorption, the choice of the life strategy, the phage gene expression and virion forming, as well as the host cell lysis will affect the outcome of phage infection shown as a phage progeny number. Thus, the observation of the changes in the lytic development efficiency of a given phage may have an explanation at one of these numerous steps. Interestingly, the lack of DksA in *E. coli* cells was also reported to affect T4 phage development, resulting in increased plaque size and a more productive infection [24], however, the mechanism involved was quite different than what we report here for phage P1*vir*. 

Here, we present the first part of our scientific story which is getting more and more interesting or “*curiouser and curiouser*”, bringing to mind “Alice’s Adventures in Wonderland” by Lewis Carroll. We would like to encourage the reader to follow this story in our accompanying paper: Virus–Host Interaction Gets *Curiouser and Curiouser.* PART II: Functional Transcriptomics of the *E. coli* DksA-Deficient Cell upon Phage P1*vir* Infection. 

## 2. Results and Discussion

### 2.1. Bacteriophage P1vir Forms Larger Plaques on a dksA Mutant Strain Than on the Wild Type Strain

Analysis of phage plaque morphology revealed that P1*vir* forms notably larger plaques when grown on a *dksA^−^* mutant when compared to the minute plaques obtained on the wild type host (Figure 1A). The plaque size was almost twice as large on the *dksA* mutant (Figure 1B) and the population of larger plaques was more abundant on the *dksA* bacterial lawn than on the wild type lawn (Figure 1C). This phenomenon raised our curiosity, because it seemed to be counterintuitive—despite the lack of DksA, its regulatory functions and its effect on RNA polymerase, P1*vir* development was not disrupted, but actually significantly enhanced. Somehow, the worse for the cell, the better for phage. We assumed that the plaque size might be a property of strain growth and not an alteration in the phage development itself, but several aspects of P1*vir* biology needed to be tested to verify this hypothesis. 

### 2.2. Analysis of P1vir Development in the Wild Type and dksA- Mutant Strains

We then decided to test several aspects of P1*vir* biology in a *dksA^−^* mutant in comparison to the wild type host, including its adsorption efficiency, phage DNA replication, expression of selected P1*vir* late genes, and the host cell membrane integrity (Figure 2A, question marks) to answer the question how P1*vir* lytic development is affected by the host DksA-deficiency.

*Adsorption of P1vir is fairly improved in a dksA host.* The very first event of P1*vir* development is its adsorption onto the host cell. Thus, the improved development of P1*vir* in *dksA* mutant could be a result of the differences in phage adsorption. To test this hypothesis we measured the efficiency of adsorption of P1*vir* phage on the wild type and *dksA* mutant hosts. We investigated phage adsorption at two different temperatures—37 °C and 0°C. Phage adsorption is moderately more efficient for the *dksA* host (Figure 2B)—approx. 77% of P1*vir* adsorb to the wild type cells, while approx. 88% adsorb to the *dksA*^-^ mutant. However this difference, while statistically significant, does not appear to be high enough to fully explain the much better P1*vir* yield. 

*DNA synthesis of P1 replicon is stimulated in a dksA mutant.* Phage DNA injected into the host cell rapidly circularizes giving a plasmid bearing two origins, *oriR* and *oriL* (see the Section 1). Replication starts at *oriR* and it is initiated by the RepA protein; then, during the lytic development replication switches to *oriL*. Here, we investigated replication of P1 DNA in the wild-type and *dksA* strains using the mini-P1 plasmid, an *oriR* replicon [25,26]. This allowed a glimpse into phage replication at the early stages of its development. The kinetics of replication of the mini-P1 plasmid were determined by pulse-labeling with 3H-thymidine and measuring the level of its incorporation into newly synthesized DNA in the course of the culture growth. We found an increased level of plasmid DNA synthesis in the *dksA^−^* mutant when comparing to the wild type strain (Figure 2C). This is especially evident at 2 h of growth corresponding to OD_600_ of 0.68 for the wild type and 0.59 for the *dksA*^-^ strain (see Appendix A, Figure A1). At this point, twice as much plasmid DNA was synthesized in the *dksA^−^* mutant than in the wild type strain. This suggests that the replication process, at least the one initiating from *oriR,* may be potentially involved in the observed phenomenon. We do not know if replication initiating from *oriL* is also affected, but an increase in *oriR* replication efficiency might cause increased number of P1*vir* plasmid copy before the replication changed to sigma model. Thus, we assume that the increased number of theta replication-derived P1 plasmid DNA leads to an increase in the number of phage genomes that will switch to sigma replication. The resulting linear phage genome molecules are then ready to be packaged and cut into newly formed virions.

*Expression of the lpa and lyz genes is downregulated in the dksA mutant.* In order to seek a link between our phenomenon of improved phage development in a DksA-deficient host and P1*vir* late gene expression, in the next stage of our investigation we employed bacteriophage P1*vir* which enters only the lytic development pathway. We assessed the level of *lpa* and *lyz* transcripts at 0, 10 and 30 min post P1*vir* infection by using real-time qPCR. The level of *lpa* transcript (encoding the late promoter activator—Lpa) significantly decreased in the DksA-deficient strain at 10 and 30 min post-infection in comparison to the wild-type strain (Figure 2D, please note the log scales). The Lpa protein activates expression of the so-called Lpa-controlled operons and is required for expression of P1*vir* late genes, e.g., lysozyme, structural proteins, and DNA packaging proteins. With some exceptions, genes involved in the late development are expressed from the late promotors and entirely or partially depend on Lpa [2]. As lysozyme is an enzyme necessary for cell lysis [17], we examined transcription of P1 *lyz* gene during the course of infection. The level of *lyz* transcript in the *dksA*^-^ mutant was decreased by two folds in comparison to the wild type, at both time points of infection (Figure 2D). Thus, it can be concluded that the DksA deficiency causes a decrease in transcription of the *lpa* gene and subsequently of the *lyz* gene, whose transcription is dependent on Lpa activation. It seems that in the *dksA*^-^ strain the lysis is less efficient. However, this more-gentle rupturing of cell membranes may lead to an increased number of virions that may be assembled before the cell disintegrates. Since we do not observe differences in the lysis time itself (discussed below and Figure 2G–J), the phenomenon could be explained by the function of efficiency, and not the function of time.

Interestingly, *E. coli* DksA deficiency was also reported to result in increased plaque size and a more productive infection by the T4 phage [24]. However, this phenomenon was accompanied by increased levels of early T4 transcripts, while in case of P1*vir* development we observe DksA-dependent modulation of late gene expression.

*DksA deficiency does not affect cell membrane integrity.* The bacterial membrane is the last barrier between the newly assembled phage particles and the environment, thus any changes in bacterial membrane composition may affect both, the adsorption of phages and the effectiveness of progeny virion release. Here, we assessed the properties of bacterial cell membranes by testing the sensitivity to organic solvents and detergents of both, the *dksA* mutant and the wild type strains (Figure 2, panels E and F). *E. coli* cells are resistant to deoxycholates due to an active efflux system [27], however, this property can be abolished by alterations in cellular membranes [28]. Thus, we monitored the efficiency of colony formation by the *dksA* mutant and the wild type strain on plates containing different concentrations of sodium deoxycholate or ethanol. The efficiency of plating was very similar for both strains at all concentrations tested (Figure 2E). We also tested cell membrane integrity by measuring activity of β-galactosidase released by the wild type and *dksA* strains. Bacteria bearing the intact *lac* operon were cultured in the presence of lactose in minimal medium and the β-galactosidase activity was measured using a modified Miller assay [29,30]. Standard assay requires 0.1 mL of chloroform to be used for each bacterial sample to ensure full release of the enzyme from the cells. We assumed that if the cell membrane of the *dksA* mutant strain was more permeable to proteins than that of the wild type strain, the release of β-galactosidase from the cells would require lower amounts of chloroform. However, comparison of the β-galactosidase activity between the wild type and *dksA* strains showed that the levels of the released enzyme are comparable between strains regardless of the chloroform amount used (Figure 2F). Interestingly, in both cases some enzymatic activity could be observed even without any chloroform added, but no differences were observed. We thus conclude that the *dksA* mutation does not affect cellular membrane integrity.

*Analysis of P1vir lytic development in a dksA host.* The most classical approach to investigate phage development is the one-step growth assay [31]. Here, we monitored release of newly synthesized phages during infection of bacterial culture, under various conditions. In this type of experiments, P1*vir* phage infection is synchronized by adding EGTA (ethylene glycol-bis(β-aminoethyl ether)-*N*,*N*,*N*,*N*-tetraacetic acid) to chelate out Ca^2+^ ions necessary for phage adsorption. Here, we found that under these conditions P1*vir* burst size in the DksA-deficient host was only moderately improved (up to 2.5× higher yield) in comparison to the wild type host (Figure 2G). This difference, however, is not sufficient to explain the observed phenomenon of larger plaques formed on the *dksA^−^* strain. Moreover, we did not observe a difference in the lysis time for the *dksA* strain—under given conditions, the P1*vir* one-step growth begins at approximately the same time for both hosts (the time point when the number of phages begins to increase is similar for each host, Figure 2G–J). 

Since the plaque size corresponds to the final effect of many cycles of phage infection and lysis, we performed a similar experiment as above but without any chelators added which allowed P1*vir* to re-infect the cells. In addition, phage development was also monitored over a longer time range. Under these conditions, phage development was measured at one-hour intervals and the results indicate a significantly higher P1*vir* burst size (from up to 6.3x) in DksA-devoid cells (Figure 2H). 

Taking into account that the P1*vir* phage development in poorly growing bacteria is typically impaired, and that during plaque formation at least some events of the infection–lysis cycle occur when bacteria have already decreased their growth rate due to entry into the stationary phase, it could be possible that the major improvement in P1*vir* development in the *dksA* strain takes place at the late stages of bacterial growth. To test this hypothesis we attempted to determine the kinetics of P1*vir* lytic growth upon infection of the host while in the stationary phase. Bacterial cells from overnight or late stage cultures (OD_600_~2.0) were infected with P1*vir* and the phage development was monitored as above. The P1*vir* development under these conditions was improved in the *dksA* host (Figure 2I), however, the difference in the burst size between the mutant and the wild type strain was similar (up to 3.3x) to that observed for bacteria infected in the exponential phase of development. We conclude that the effect of DksA-deficiency on P1*vir* development is not more pronounced for slowly growing bacteria than for those in the log phase of growth. However, the moderately improved lytic development observed in the prolonged one-step growth experiment may at least partially account for the observed phenomenon of larger P1*vir* plaque sizes formed on the *dksA* mutant strain.

*Overproduction of DksA does not affect P1vir lytic development.* As the observed consequence of DksA-deficiency was a notable improvement in lytic growth of P1*vir*, manifested by larger plaques and a higher phage yield, it was plausible to assume that a DksA excess may result in an opposite effect. We thus assessed the kinetics of P1*vir* lytic development upon overproduction of DksA from the pBR322-derived pJK537 plasmid [18] bearing the *dksA* gene under its natural promoter (*dksA* present on this plasmid results in about 20-fold increase in cellular level of DksA [20]). However, we found quite unexpectedly that excess DksA did not inhibit P1*vir* growth, neither affecting the burst size, nor the time when lysis occurred. In fact, P1*vir* growth was very similar in the *dksA*-proficient and deficient hosts bearing pJK537 (Figure 2J). Observation that DksA excess does not influence P1*vir* development, while DksA deficiency notably improves it, seems quite contradictory. One plausible explanation is that the influence of DksA on phage lytic growth is already at its fullest when DksA is present at its natural level and the protein excess does not make this effect any stronger.

*Summary of this section.* Three different steps of P1*vir* development seem to be affected by the host DksA deficiency. P1*vir* adsorption is moderately improved, but it is unlikely that this minor improvement is solely responsible for increased P1*vir* phage development. However, combined with improved phage DNA replication and reduced host cell lysis efficiency (which in itself is a counterintuitive idea), these minor improvements might together account for the observed phenomenon. Therefore, we next decided to study in detail the regulation of P1*vir* phage development in the *dksA* mutant host.

### 2.3. Regulation of P1vir and P1wt Phage Development in a dksA^−^ Host

In general, the P1 phage development is maintained by a complicated interplay of components of three immunity regions—*imm*C, *imm*T and *imm*I (for details, see Introduction and Figure 3A,B). Lack of DksA, one of the major cellular regulators, complicates this regulation even more. In addition, P1*vir* phage ability to only enter the lytic pathway is caused by spontaneous but unknown mutations located in the *imm*I region [2,9]. Therefore, we addressed the following questions: (i) What is the expression level of C1 (the main repressor of lysogeny) during P1*vir* infection of the *dksA* mutant and wild type strains? (ii) Is the *imm*I region of P1*vir* really harboring any mutations, as it was claimed back in the 1980s, and if so then how can these changes be interpreted in the light of improved P1*vir* development in a *dksA^−^* host? (iii) Is the P1*wt* (the temperate phage) development also affected in the *dksA^−^* strain?

*Expression of C1, the main lytic repressor, is downregulated in a dksA^−^ strain*. The level of *c*1 gene transcript is significantly decreased at 10 and 30 min after P1*vir* infection of the *dksA* mutant in comparison to the wild type strain (Figure 3C). This suggests that the host’s DksA protein may affect expression of the P1 phage main lytic repressor—C1. On one hand, this might be linked to DksA being a transcriptional factor exerting many pleiotropic effects and thus a link between decreased *c*1 expression and improved P1 phage development in a DksA-deficient host might be indirect. On the other hand, low level of C1 might be one of the “minor improvements” which lead to enhanced phage development. Because C1 repressor regulates expression of lysogeny maintaining genes by binding to C1-operator sites, we hypothesize that lower level of C1 may lead to less effective repression of some of these genes, thus leading to a sort of “expression leakage”. In this scenario, lysogeny genes may compete for access to the transcription machinery with lytic genes. Ultimately, this may result in a less efficient or less abrupt lysis and therefore, under favoring circumstances, the host cell may be able to produce more phage particles due to less efficient lysis in comparison to the wild type strain.

*Mutations located in the imm*I *region lead to the lytic pathway in P1vir development*. The P1 phage’s improved lytic development in a *dksA* mutant that we describe here was observed for its virulent variant— the P1*vir* strain. It needs to be mentioned that P1*vir* was not engineered—it was isolated from nature by Sakar in 1960 [32]. This virulent variant of P1 has been used in almost every molecular biology laboratory for strain construction via generalized transduction. However, not much is known about the actual molecular mechanism of its natural virulence. Till now, it has been only reported that this phage was unable to enter the lysogenic cycle due to a mutation located in the *imm*I region [9]. We thus performed DNA sequencing of this region and found three nucleotide substitutions in P1*vir* in comparison to the P1*wt* genomic sequence (Figure 3F). One of these mutations is located in the P2c4 promoter region, and thus it could affect expression of the C4 antisense RNA. The second mutation is located near the C4-antisense-RNA binding site, and thus it could affect the C4 binding ability and therefore alter expression of the rest of the *imm*I region that encodes the *icd* gene (its expression suppresses cell divisions) and *ant1/2* genes (encoding the secondary anti-repressor Ant which is also the only anti-repressor that can be expressed in the presence of C1). The third mutation is located in the PkilA promoter region, and thus it could affect expression of *kilA*, which encodes a protein whose overexpression leads to cell death [2]. Still, an open questions remains whether all of these mutations are necessary to cause the permanent virulence of P1*vir*.

*The P1wt lytic development in a dksA host*. We have tested the efficiency of P1*wt* lysogenization and found no statistically significant differences between lysogenization frequency in a *dksA* mutant and the wild type strain (Figure 3D). We thus hypothesize that the main mechanism leading to more efficient P1*vir* development in the *dksA* mutant solely involves the lytic pathway regulation. Therefore, we analyzed the P1*wt* plaque morphology. Visually, it was impossible to determine if there was any difference between the plaques formed on the wile type or *dksA* strains (Figure 3E, photographs). However, detailed image analysis showed that plaques formed on the *dksA* host are slightly but significantly larger than those formed on the wild type cells (Figure 3E, plaque size graph). In addition, the population of larger plaques is more abundant on the *dksA* bacterial lawn than on the wild type lawn (Figure 3E, plaque size distribution graphs). We then analyzed the clarity of plaques with the integrated density values for each plaque provided by image analysis. We found that P1*wt* forms a population of significantly clearer plaques on the *dksA* strain than on the wild type strain (Figure 3E, plaque clarity graph). The clear plaques are indicators of lytic-only development and the more opaque plaques reveal that part of the phage population turns into lysogens allowing bacterial growth. It needs to be said that although these changes are not very prominent, they are statistically significant and thus they must be taken into account.

*Summary of this section:* In general, P1 shows great flexibility in host selection, which is mainly due its complex regulation. The lysis or lysogeny decision is taken as a result of a complex interplay of three immunity regions—*imm*C, *imm*T and *imm*I. Permanent imbalance in either *c*4 gene expression or translation blockage of *ant1/2* expression favors the lytic development [1,2]. As was reported earlier, mutations located in the *imm*I region of P1*vir* lead to the lytic-only development pathway. Moreover, changes in regulation of *kilA* expression can directly affect the host cell, e.g., an increase in the KilA protein level leads to bacterial cell death. At this point it is impossible to say which of the mutations that we detected in the P1*vir imm*I region are essential for the lytic-only pathway. However, it can be assumed that a small but permanent imbalance in the interplay of phage immunity factors leads to P1 virulence.

Lack of DksA enhances lytic development of P1*vir*, and expression of C1, the main lytic repressor, is downregulated in the *dksA* strain during P1*vir* infection. This suggests that P1*vir* may be balancing between lysis and lysogeny although finally it is able to enter the lytic pathway only. Analysis of P1*wt* lytic development in the *dksA* host also shows that it is slightly but significantly improved. This suggests that the lack of DksA protein, one of the major global regulators, affects both phage strains—P1*wt* and P1*vir*—but in case of P1*vir* the effect is much more prominent.

## 3. Materials and Methods

### 3.1. Bacterial Strains and Phages

Bacterial strains used in this work: CF1648—MG1655 wild type [33]; CF9240—MG1655 *dksA*::Tn10 [34]; CF7991—MG1655 wild type [34]; CF7991—*dksA*::Tn5 [18]; AEC238—CF10440 *dksA*::*tet* (this work). Cultures were routinely grown at 30 °C or 37 °C in lysogeny broth (LB) supplemented with antibiotics as needed (ampicillin, 50 µg/mL, tetracycline 15 µg/mL, kanamycin, 50 µg/mL). Phages: P1*vir* and P1*wt* from the collection of the Department of Bacterial Molecular Genetics, University of Gdansk, Poland.

### 3.2. Plaque Morphology

Bacteriophage P1*vir* and P1*wt* plaque morphology was analyzed in the wild-type and *dksA* mutant cells. Serial dilutions of P1 phage lysate were done in the TM buffer (10 mM Tris-HCl pH 7.4; 10 mM MgCl_2_). Next, 200 µL of over-night cultures of tested strains were mixed with P1 diluted lysate (10^2^–10^3^ phages per each tube were added). Next, 3 mL of warm top agar was added (1% peptone, 0.5% NaCl, 0.7% bacteriological agar, 10 mM MgCl_2_), quickly mixed and poured onto freshly prepared LB agar plates. Plates were incubated over-night at 37 °C and scanned at 600 dpi for image acquisition. Morphology of plaques was analyzed using Photoshop SC3 image analysis tools. The following parameters were obtained: plaque Area (for P1*vir* and P1*wt*) and Median Gray Value of each plaque (for P1*wt*); the latter parameter was used to calculate the Relative Plaque Clarity. Relative Plaque Clarity was equal to 1/Integrated Density; Integrated Density was calculated as Median Gray Value divided by Area.

### 3.3. Efficiency of Bacteriophage P1 Adsorption

The efficiency of phage adsorption was assessed as described previously [35]. Briefly, bacteria were infected with phage lysate at MOI = 1. After 10 min incubation at the indicated temperature (ice-bath or 37 °C), cells were pelleted by centrifugation and the number of unadsorbed phages was estimated by plating dilutions of the supernatant on a wild type (MG1655) strain. The efficiency of adsorption was calculated from the difference between the total number of input phages and the number of unadsorbed phages.

### 3.4. Kinetics of Mini-P1 DNA Replication

The efficiency of DNA synthesis was determined as described previously [36]. Briefly, overnight cultures of bacteria bearing plasmids were diluted 100-fold into fresh LB medium, and cultivation was continued at 37 °C until OD_600_ = 0.10 ± 0.01. Next, culture samples were withdrawn (1 mL) and added to 2 mL-tubes containing the radioactive precursor ([^3^H]thymidine at the final concentration of 10 µCi/mL) and incubated for exactly 4 min at 37 °C, then chilled on ice and pelleted by centrifugation (10 min; 4000× *g*; 4 °C). Plasmid DNA was isolated using the GenElute™ Plasmid Miniprep Kit (Sigma-Aldrich, Saint Louis, MO, USA), according to the manufacturer’s protocol, and radioactivity (shown as counts per minute—CPM) of each sample was measured with a MicroBeta2 scintillation counter (Perkin-Elmer, Waltham, MA, USA). Efficiency of plasmid DNA synthesis was calculated as CPM of isolated plasmid DNA divided by the optical density of culture at a given timepoint. 

### 3.5. Membrane Integrity Assays 

The β-galactosidase activity was measured as described by Miller [29]. A modified protocol was used in order to assess permeability of membranes by measuring the amount of the enzyme released from bacterial cells in the presence of various amounts of chloroform. To assess bacterial growth on ethanol and deoxycholate, bacterial cultures grown in LB medium were diluted and spread on LB plates containing indicated concentrations of sodium deoxycholate or ethanol. Plates were incubated overnight at 37 °C and the number of colonies was counted.

### 3.6. Kinetics of Phage P1 Lytic Growth

Lytic development of bacteriophage P1*vir* was investigated by one-step growth experiments [31]. Briefly, exponential or stationary phase bacteria were infected with the P1*vir* phage and allowed to adsorb for 10 min in the presence of 3 M NaN_3_ in order to obtain synchronized infection. Bacteria were then pelleted and resuspended in 1/10 initial volume of LB medium containing 3 M NaN_3_ and 10 mM EGTA (ethylene glycol-bis(2-aminoethylether)-*N*,*N*,*N*′,*N*′-tetraacetic acid) to prevent re-adsorption. Next, the cells were diluted 10-fold with a prewarmed medium (without NaN_3_) and cultured with shaking in a water bath. The number of infective centers was estimated by plating samples withdrawn during the first 10 minutes after dilution, on the indicator strain MG1655. The number of phage progeny was calculated by plating samples taken at indicated times and the burst size was estimated as the ratio between the number of phage progeny normalized to the number of infective centers. For DksA overproduction, pJK537 plasmid (a pBR322 derivative) was used [18].

### 3.7. Efficiency of Lysogenization

Bacterial wild type and *dksA* strains were cultured in LB broth containing 10 mM CaCl_2_ to OD_600_ = 0.2, at 30 °C with shaking. Next, 2 × 1 mL of each culture was withdrawn and added to two separate 1.5-mL-tubes. To one of the tubes, 30 µL of P1*wt* lysate (6.7 × 10^9^ phages) was added (the other tube was treated as control). All tubes were incubated at 30 °C for 30 min with shaking. Next samples were pelleted by centrifugation (4000× *g*, 10 min) and resuspended in fresh LB broth on ice. Serial dilutions were made and each was spread on LB agar plates, as well as on LB agar plates supplemented with chloramphenicol (P1*w*t is bearing a chloramphenicol resistance gene). Efficiency of lysogenization was calculated as the number of lysogen forming units (plate with chloramphenicol) in comparison to the total number of colony forming units (plate without any antibiotic) in each sample. 

### 3.8. Sequencing of the C4-ant1/2 Region

The *C4-ant1/2* region of the P1*vir* phage was PCR amplified using primers flanking the region of interest (*c4-ant2_1*—CCT CAC CTG CCT TAT AAC, *c4-ant2_2*—CCA GCT CGA AAT GGT GAT), P1*vir* phage genome—previously directly isolated from the lysate using the GenElute Bacterial Genomic DNA Kit (Sigma-Aldrich, Darmstadt, Germany), and the WALK polymerase (A&A Biotechmology, Gdańsk, Poland) containing the high-fidelity *Pwo* DNA polymerase dedicated for amplification of long fragments. DNA sequencing was performed by Macrogen Europe Inc., Amsterdam, the Netherlands. Sequencing data were analyzed using the NCBI GenBank reference number NC_005856.1 (Enterobacteria phage P1, complete genome) as the reference sequence.

### 3.9. Total RNA Isolation

Bacterial wild type and *dksA* strains were cultured in LB broth containing 10 mM CaCl_2_ to OD_600_ = 0.2, at 37 °C with shaking. Next, P1*vir* was added at final MOI = 10. One-mL samples were immediately withdrawn to previously prepared 1.5 mL tubes containing 250 µL of aqua-phenol (5% phenol in 96% ethanol). Samples were withdrawn at 0, 10 and 30 min after P1*vir* infection. Next, RNA isolation was performed using RNeasy Mini Kit (Qiagen, Hilden, Germany) according the manufacturer’s protocol. The elution step was performed twice using 50 µL of RNase-free water on the same column to increase the RNA yields. Thus obtained RNA was digested with a DNase using TURBO DNA-free™ Kit (Ambion, Carlsbad, CA, USA), according to the manufacturer’s protocol. Samples were stored at −80 °C. The purity and integrity of RNA was assessed using the Bioanalyzer 2100 (Agilent Technologies, Santa Clara, CA, USA) and Agilent RNA 6000 Nano Kit, according to the manufacturer’s protocol.

### 3.10. Real-Time qPCR

Real-time quantitative PCR (RT-qPCR) analysis was carried out with *c1*, *lpa*, and *lyz* gene-specific primers (listed below) using SYBRGreen. A housekeeping 16S DNA gene was used as an internal control. Reverse transcription was performed using EvoScript Universal cDNA Master (Roche) according to the manufacturer’s protocol. Briefly, synthesis of cDNA was performed in a total volume of 20 µL; the reaction contained 1 µg of total RNA, 4 µM random hexamers and 1 mM dNTP Mix. The cDNA synthesis conditions were as follows: 42 °C for 15 min, 85 °C for 5 min, 65 °C for 15 min, 4 °C for 15 min. The RT-qPCR amplification was performed in a total volume of 10 µL using the 1X SYBR Green I Master Mix (Applied Biosystems), 0.5 µM primers and 1 µL of the cDNA template. Reactions were carried out in the LightCycler^®^ 480 Instrument II (Roche Diagnostics Ltd., Rotkreuz, Switzerland) with the following amplification program: pre-incubation at 95 °C for 5 min, followed by 45 cycles of 95 °C for 10 s, 60 °C for 10 s and 72 °C for 10 s. Melting curve analysis was conducted by ramping the temperature from 65 °C to 95 °C for 1 min. Primer sequences: *lpa.1*—AAA GGA ATA CCC GCA TCT GG); *lpa.2*—CAC AGC GAT AGC CTT TAA GC; *16S.1*—CTC CTA CGG GAG GCA GCAG); *16S.2*—GWA TTA CCG CGG CKG CTG); *lyz.1*—TAT TAC CGA TCC CGT CAGTC; *lyz.2*—GCT ATC GCG GTG ATG ATT AC; *C1.1*—GGC CAG CGG TCA TAT TGA CG; *C1.2*—TAA TGG CGC GGG AAC TGG AG.

## 4. Conclusions

The development of P1 phage is well studied, but to our surprise we observed that P1*vir* lytic development is improved in the absence of the DksA protein. Intrigued by this observation, we investigated several aspects of P1*vir* biology in an attempt to understand the basis of this phenomenon but did not find a single genetic switch that could explain it. Instead, we found several minor improvements of phage development in the *dksA* mutant. In the absence of the DksA protein, slightly more P1*vir* phages can adsorb to bacterial cells, the phage DNA replication is more efficient and the cell lysis seems to be less efficient or less abrupt. Therefore, the DksA-deficient host cell may be able to produce more phage particles in comparison to the wild type strain. We think that P1*vir* may be balancing between lysis and lysogeny although finally it is able to enter the lytic pathway only. Further studies, documenting gene expression changes in P1vir and its *E.coli dksA*^-^ host upon infection seem to strengthen this hypothesis (see the accompanying paper: Virus–Host Interaction Gets *Curiouser and Curiouser.* PART II: Functional Transcriptomics of the *E. coli* DksA-Deficient Cell upon Phage P1vir Infection). Of note, we also identified here three nucleotide substitutions located in the P1*vir imm*I region which may be responsible for lytic-only development of P1*vir*, but the question whether all of them are necessary to the permanent virulence of P1*vir* remains open. 

## Figures and Tables

**Figure 1 ijms-22-05890-f001:**
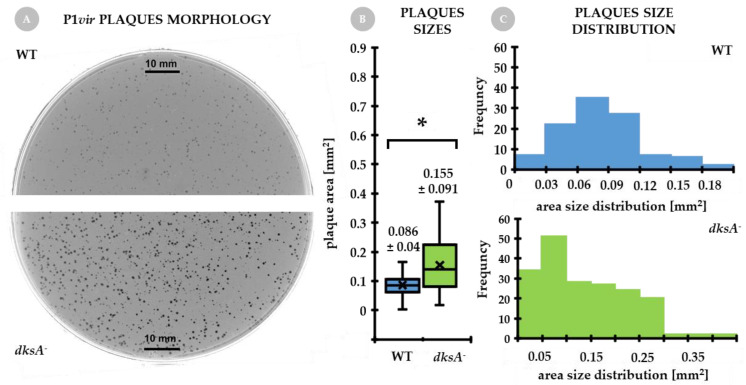
Bacteriophage P1*vir* plaques morphology. The same number of phage particles (10^2^–10^3^ phages per plate) was added to the top agar containing wild type or *dksA^−^* cells. (**A**) Representative pictures of plaque morphology obtained for the wild type and *dks*A mutant strains. (**B**) Calculated P1*vir* plaque sizes when developing on either strain (results are presented as mean ± SD, whiskers show max and min values, black line across boxes represents the median). (**C**) Distribution of plaque sizes for both infected strains. Note different values on the x-axis for each strain. Data were collected from at least 100 plaques obtained in at least three independent biological replicates per strain. Statistical significance was tested with the Mann–Whitney U test; (*) marks significant differences with *p*-value ≤ 0.05.

**Figure 2 ijms-22-05890-f002:**
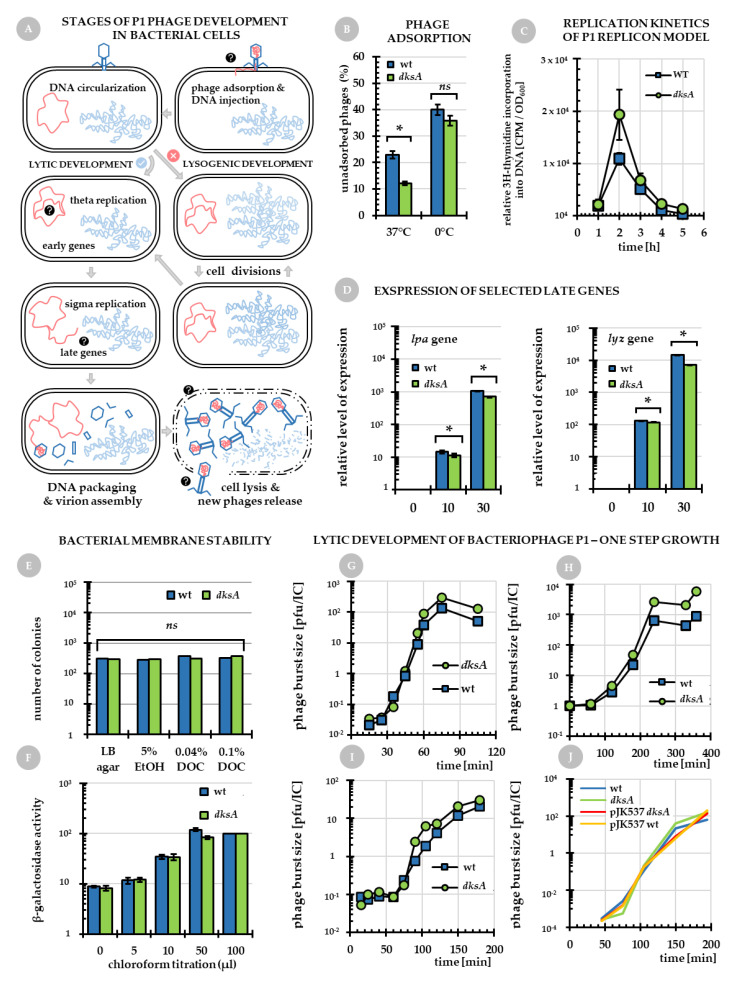
Analysis of selected aspects of P1*vir* phage biology in different hosts. (**A**) General overview of P1 phage development in *E. coli*. Question marks denote hypothetical points in phage/host biology where changes may lead to the observed phenomenon of improved P1*vir* development in a *dksA* mutant vs. the wild type strain. These points were experimentally verified and the results are shown in following panels. (**B**) Efficiency of P1 phage adsorption on wild type (blue columns) and *dksA* (green columns) strains presented as percentage of unadsorbed phages at indicated temperatures. (**C**) The efficiency of pSP102 DNA synthesis, a bacteriophage P1 replicon (the experiment was performed during the exponential phase of growth; compare Figure A1). Panel **(D)** shows expression levels of phage genes encoding late promoter activator (*lpa*) and lysozyme (*lyz*) during P1*vir* infection of the wild type and *dksA* mutant cells. Gene expression at 10 min and 30 min of infection was compared to time 0 in the corresponding strains (note the log expression scales). (**E**,**F**) Data obtained for the host membrane stability tests. (**E**) Colony formation efficiency of the wild type and *dksA* strains. Bacteria were cultured in LB and then were diluted and plated on LB agar plates containing indicated concentrations of ethanol (EtOH) or sodium deoxycholate (DOC). (**F**) The relative β-galactosidase activity of the wild type and *dksA* strains treated with different amounts of chloroform. β-galactosidase activity that was assessed in samples treated with 100 µL of chloroform was set as 100% for each strain. (**G**–**J**) show lytic development of bacteriophage P1 in the wild type (green circles) and *dksA* (blue squares) hosts at 37 °C. The phage burst size was calculated as plaque forming units per infective center (pfu/IC). Bacteria were cultured after infection in LB containing 10 mM EGTA—panel (**G**). Cells were diluted in LB and the experiment was carried out for a longer period of time (**H**). Bacteria from overnight cultures were used for infection (**I**). (**J**) Effect of DksA overproduction. P1*vir* development in the wild type strain harboring pJK537 plasmid is represented by a yellow line and *dksA* mutant harboring pJK537 by a red line. Blue line indicates the wild type strain and green line determines the *dksA* mutant, not harboring plasmids. For all panels: blue squares or columns represent the wild type cells and green circles or columns represent the *dksA* mutant. Presented results are mean values obtained from at least three independent experiments. Error bars represent SD values. Statistical significance was estimated with *t*-test; asterisk (*) marks *p* < 0.05 and*(ns*) stands for not significant.

**Figure 3 ijms-22-05890-f003:**
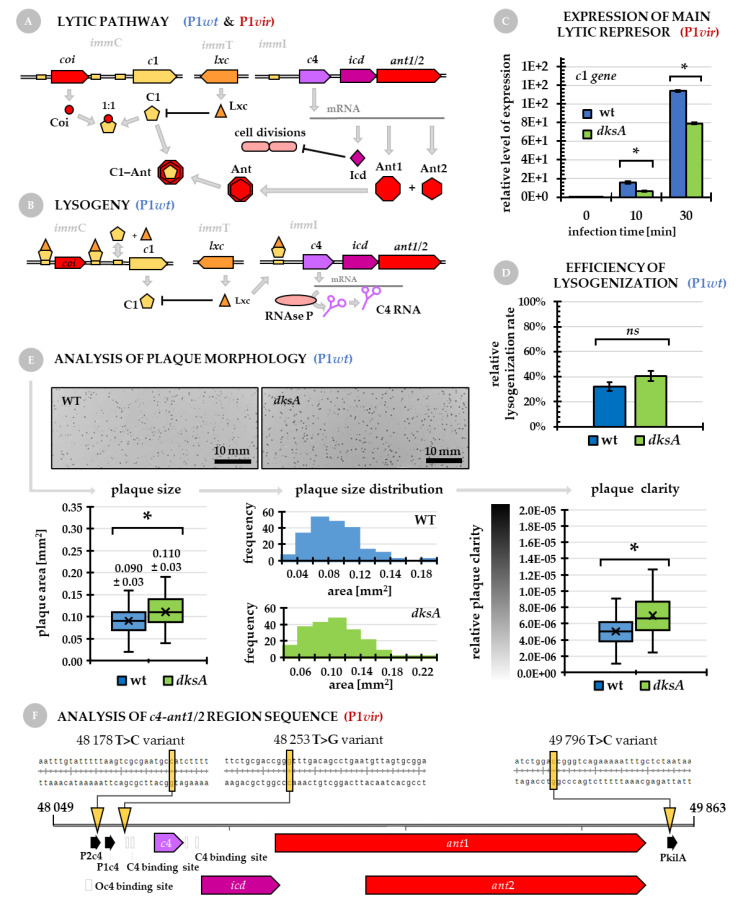
Regulation of P1*vir* and P1*wt* phage development. (**A**,**B**) Molecular regulation of the lytic pathway (P1*wt* and P1*vir*) and lysogeny maintenance (P1*wt*). In the lytic pathway, Coi, an anti-repressor, binds to C1, the primary repressor of lytic functions. The secondary anti-repressor Ant (Ant1-Ant2 dimer) binds to the C1 repressor lowering its level. At the same time, increasing concentrations of the Lxc protein expressed from the *imm*T region lead to suppression of *c*1 gene expression. Next, genes in the *imm*I region that are not blocked by the C4 antisense RNA, are expressed (compare panels (**A**) and (**B**))—the Icd protein suppresses cell division and Ant is a secondary anti-repressor. (**C**) Regulation of *c*1, the main lytic repressor gene, upon P1*vir* infection. l (**D**) Efficiency of P1*wt* lysogen formation in the wild type and *dksA* mutant strains. (**E**) Results of P1*wt* phage plaque morphology obtained for the wild type and *dksA* host strains. Photographs present general overview of plaques formed; graphs present plaque size (values are the mean area ± SD, whiskers show max and min, line across the boxes represents median), plaque size distribution, and relative plaque clarity (calculated as a reciprocal of integrated density values, which is the plaque median gray value divided by the plaque area). Asterisk (*) marks *p* < 0.05 and *(ns*) stands for not significant (**F**) Location of spontaneous mutations in the c4-ant1/2 region (illustrating how P1*wt* became P1*vir*) revealed by sequencing. Two substitutions in P1*vir* were found to be located in the P2c4 promoter region, near the antisense-C4-RNA binding site, and one in the PkilA promoter region.

## Data Availability

Data is contained within the article or Appendix A.

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
