# Peer review of "Virus–Host Interaction Gets Curiouser and Curiouser. PART I: Phage P1vir Enhanced Development in an E. coli DksA-Deficient Cell"

_ijms, 2021, doi:10.3390/ijms22115890_

Round 1

Reviewer 1 Report

Revision of manuscript ijms-1225313

Dear Authors,

Your manuscript entitled “Virus-Host Interaction Gets Curiouser and Curiouser. PART I: Phage P1vir Enhanced Development in an E. coli DksA-Deficient Cell” describes an interesting study of basic research. Authors fully and well explored the particular interaction of P1vir phage and a DksA-deficient host. Different questions were placed and solved with a multistep approach.

The work is well planned and presented. Results are well and clearly presented and fully discussed.

I appreciate the style of the manuscript proposed by Authors.

Sincerely

The Reviewer

Author Response

Dear Reviewer,

Many thanks for the revision. We believe a good story is as important as reliable data. That is why we are glad that our work has been noticed.

Sincerely,

The Authors

Reviewer 2 Report

Very interesting paper dealing both with recent developments of research and historical aspects of the topic (confirmed by many "old" references cited). It is clearly presented and easy to follow.

I do not think that the figure A1 is really important. The measure of growth by OD is not very sensitive (as extracellular polysaccharides for instance can also increase the trouble without an increase of bacteria) and the differences are not significant. This could be mentionned in the text.

Author Response

Dear Reviewer,

Thank you for your revision.

As for Figure A1, we know that the data may not be crucial but it was included for two reasons: (i) to show in which phase of growth are bacteria during the experiment shown on Figure 2C; (ii) to show the general growth rate of bacterial strains – as this is mainly the first phenotype observation that is helpful in day-to-day experiments with the wild type and dksA mutant strains. Despite the very small difference shown on figure A1, dksA mutant grows slower and does not reach the optical density as the wild type strain. We are well aware that measuring the OD it is not very exact method, however, we wanted to show that the replication kinetics of P1 replicon model (shown on Figure 2C) was performed during the exponential phase of growth. Thus, we decided to improve Figure A1 and show the data using the semi-logarithmic scale (lines 566-567) and we included this information in the description of Figure 2C (lines 150-151). We also slightly rephased the description of Figure A1 (lines 568-571) and included the information that it is an additional information for Figure 2C.

Sincerely,

The Authors

Reviewer 3 Report

The research manuscript provides novel data on the phage Pvir biology. The authors analyzed in detail, using genetic and microbiological approaches, the P1 phage development in an E. coli DksA-deficient cell. The results showed that P1vir lytic development is improved in the absence of the DksA protein, resulting in increased plaque size and more productive infection. These data are contributing to the knowledge of P1 phage biology, which is an important model for the study of many processes in molecular biology including aspects of phage host interaction.

Author Response

Dear Reviewer,

We would like to thank you for the revision. We are glad that our work has been appreciated. 

Sincerely,

The Authors